# The relationship between artistic skills and academic engagement among artistically gifted non-specialist university students

**Ghozeail Abdulaziz Abdullah Aldhorman**⬤*

Art Education, Department of Home Economics, College of Education, Prince Sattam bin Abdulaziz University, Al Kharj, KSA

* g.aldhorman@psau.edu.sa

## Abstract

### Background

This study aimed to investigate the level of artistic skills and academic engagement among artistically gifted non-specialist students at Prince Sattam bin Abdulaziz University, Saudi Arabia. It also attempted to identify the relationship between students' artistic skills and academic engagement. The study followed a mixed methods design where both quantitative and qualitative data was collected and analyzed. The participating students (102 students enrolled in the university's Gifted Care Unit) completed an author-developed questionnaire probing their artistic skills (cognitive, performance, and emotional) and academic engagement (vigor, dedication, and absorption). In addition to its 30 Likert scale items, the questionnaire had an open-ended question about ways to promote artistic skills from the perspective of the students. Furthermore, the author electronically interviewed 14 members of the Gifted Care Unit and got them to respond to the same open-ended question in the students' questionnaire, i.e., what are your suggestions for promoting students' artistic skills. The quantitative data analysis via descriptive statistics revealed that the students had high artistic skills and academic engagement. Pearson correlations revealed a significant positive relationship between artistic skills and academic engagement. Standard regression analysis produced a significant model where artistic skills explained 11% of variance in students' academic engagement. Of the individual dimensions of artistic skills, only performance skills explained 13% of students' academic engagement. The analysis of students' and members' responses to the open-ended question revealed some shared themes, e.g., the students' need for more specialized training, contact with professional experiences, support from significant others, and creativity-stimulating competitiveness.. A number of implications are offered.

**Data availability statement:** All relevant data are within the paper and its Supporting Information files.

**Funding:** This study is supported via funding from Prince Sattam bin Abdulaziz University project number (PSAU/2025/R/1446).

**Competing interests:** The authors have declared that no competing interests exist.

## Conclusion

The important finding in this study is that ASs, especially performance skills, contribute to students' AE. This makes it important for specialized entities like the Gifted Care Unit to exert more efforts to identify artistically gifted students in all university programs. These efforts can identify students who are aware of their artistic skills and students who have artistic skills but are unaware of them. The art culture should be spread all over the university to attract more students to join the Gifted Care Unit. The artistically gifted students should be provided with specialized training, contact with outside sources of artistic experience, an art-nurturing environment, incentives to be more creative, and support, especially spiritual support.

## Introduction

Supporting gifted students is of great importance, and through a number of tailored educational experiences, these students can excel in their academic endeavors, develop their talents, and promote their innovation in areas such as arts. Fine arts represent a vital human activity. In the current era, many different arts have appeared that suit the inclinations of many young people, e.g., drawing, sculpture, ceramics, photography, and printing. These arts have attracted young people to learn them, and thus a generation of young people who are gifted in various fine arts emerged.

Giftedness is "an innate aptitude that the individual is born with" [1]. Gifted students are characterized by many distinctive abilities. They are distinguished by "a multiplicity and diversity of interests, a breadth of knowledge, and a high degree of excellence in reading, use of language, science, literature, and arts" [2]. Gifted students are distinguished by high academic achievement. They also "demonstrate special excellence and creativity in artistic fields such as drawing and they can be identified by specialists" [3]. The artistically gifted gifted non-specialist university student in this study is defined as a university student who studies in specializations other than fine arts and is nominated as gifted in fine arts after passing a test and an interview. AlDharman (2021) reported a relationship between artistic skills (hereafter ASs) and talent [4]. Skills refer to the ability to translate ideas into tangible meanings with flexibility, originality, fluency, and economy of time and effort. Artistically gifted individuals have skills that they can adapt to different situations.

Academics and decision-makers in educational institutions have emphasized the need to explore the academic engagement (hereafter AE) of university students as it is the key to addressing many academic, personal, social, and psychological problems [5,6]. AE is defined as "the extent to which the student adapts academically, educationally, psychologically, socially, and emotionally to the university life" [7]. AE is affected by a number of external factors, such as the academic climate, the definition of the student's role and academic status, feedback from teachers, and the teaching methods used. It is also affected by the internal circumstances, e.g., interactions within the classroom, students' perceptions of their educational environment, and their feeling of

effective participation, self-efficacy, motivation, enthusiasm, and passion for learning [8]. Many studies, e.g., [9,10] reported that improving these factors increases satisfaction with the educational process and thus improves the degree of AE.

There is a need to study the ASs of artistically gifted non-specialist Saudi university students and explore if possessing ASs can have a positive reflection on these students' AE. More specifically, this study addressed the following questions:

1- What is the level of ASs among artistically gifted non-specialist students?

2- What is the level of AE among artistically gifted non-specialist students?

3- What is the nature of the relationship between ASs and AE among artistically gifted non-specialist students?

4- What are the ways to raise the level of ASs among artistically gifted non-specialist students from the perspective of students and members of the Council of the Gifted Care Unit?

## Research significance

This study can inform researchers, educators, social workers, and those responsible for artistically gifted students in Saudi universities of how to encourage students and invest their artistic energies for their benefit and the benefit of their community.

It sheds light on the level of Saudi university students' ASs, which can illuminate the identification and nurturing of gifted students.

It is expected to identify the relationship between ASs and AE, which has not been researched to the best of the author's knowledge.

It can provide a tool to measure the level of ASs and AE that can be used by researchers in Saudi Arabia.

## Literature review

**Artistic skills.** Hanna [11] defines ASs as "artistic practices that require the use of arm, hand, and finger movements. This practice is characterized by speed and accuracy, as well as by efficiency and experience." Hala and Xhomara [12] define them as "specialized knowledge and experience that support the performance of specific tasks in the work environment. They include the techniques, tools, and procedures a person needs to do their job effectively". According to Abdelhafeez [13], ASs refer to "the accuracy in using numbers and tools, or care in performance, and the ability to mix colors, find relationships, and formulate composition, all of which have two aspects, one of which is innovative and the second is memorized as a result of training."

Mahmoud et al. [14] Suggest that students' ASs fall within three basic dimensions: cognitive skills, performance skills, and emotional skills. Al-Awadi and Al-Kharusi [3] believe that the general basic ASs include three basic dimensions. The first dimension is plastic skills, which are concerned with the physical aspect of ASs. The second dimension is expressive skills, which are concerned with the emotional aspect of ASs, i.e., the student's feelings and emotions. Technical skills constitute the third dimension, which is concerned with the performance aspect of ASs.

Because the development of ASs is an important outcome for university students, many studies addressed the development of ASs among university students. Ahmed [15] examined the development of ASs through ceramic ornaments among secondary school students as an introduction to creating a small project. The study followed the semi-experimental method. It concluded that it is possible to develop the artistic expression of secondary school students and enrich their innovative abilities through ceramic formation, which could qualify them to start small businesses.

Ibrahim [16] used the aesthetic formulations of batik using the two methods of performance immersion and engraving on wax to produce luminous print commentary to develop the ASs of a sample of 10 students at the Faculty of Specific Education, Ain Shams University. The results revealed that the problem of producing the design with clear lines using batik was solved by engraving on wax instead of using brush painting to achieve tactile and aesthetic values and find contemporary plastic solutions.

Salama [17] aimed to design an instructional workshop strategy as an introduction to developing the ASs of non-specialists, and to design an observation checklist to monitor the strengths and weaknesses of the workshop with the aim of raising the artistic taste and competence of non-specialists. The study sample consisted of 45 female students from five colleges at Suez University. The results indicated that the instructional workshop contributed to raising the level of students' ASs, although they came from different specializations.

Al-Ghamdi [18] targeted developing the artistic imagination skills of intermediate school students through stories. The study followed the semi-experimental method. Data was collected from 30 students using the Artistic Imagination Skills Inventory. The results revealed that there were statistically significant differences between the participants' pre- and post-test mean scores on the inventory's individual dimensions and the total score of the artistic imagination skills in favor of the post-test, thus supporting the effectiveness of using stories in art education.

Hanna [11] explored the effectiveness of infographics in developing some ASs among second-graders of the preparatory school. To achieve this goal, a teacher's guide, student worksheets, and an observation checklist were developed to clarify and assess the effectiveness of using infographics in teaching art to the study population. The results supported the effectiveness of infographics in developing some of the participants" ASs.

Saleh and Al-Bashir [19] investigated the characteristics of the deaf and assessed the level of their cognitive and creative abilities in artistic applications. The study followed the descriptive analytical method, where a questionnaire was administered to a sample of deaf students at the bachelor and diploma levels. The study concluded that the participants displayed superior cognitive and sensory abilities, creative skills, and artistic imagination for good expression.

Khalil [20] revealed the plastic treatments of plastic consumables to develop the artistic skill of 20 art students at the Faculty of Art Education in Zamalek. The results revealed that the construction and enrichment of the artwork and the development of the artistic skill of the participants were achieved through multiple plastic treatments and methods of constructing and enriching the artwork. The participants' artistic skill in assembling and constructing the artwork developed. The study recommended that art teaching programs should be developed to promote the students' ASs.

**Academic engagement.** Schaufeli et al. [21] define AE as a "state of mind that is characterized by vigor, dedication, and absorption. Rather than a momentary and specific state, engagement refers to a more persistent and pervasive emotional cognitive state that is not focused on any particular object, event, individual, or behavior". Conner [22] suggests that engagement is similar to terms such as activity, participation, interest, motivation, and effort. Engagement is commitment to an activity, and it mediates the relationship between students and activities. Martin and Torres [23] suggest that AE is linked to learners' interactions with various educational components such as the community, university staff, classmates, teachers, and courses. Martin, et al. [24] define AE as "a psychological state characterized by a student's sense of belonging, attributing value to education, and participating in school, learning, studying, and curricular activities. AE is particularly characterized by vigor, dedication, and absorption". Through a two sample confirmatory factor analytic approach, Schaufeli et al. [21] extracted a model of AE comprising vigor, dedication, and absorption. Initially, engagement was conceived of as work-related, but it has recently been expanded to include AE. According to Schaufeli et al. [21], the student, like the employee, participates in arranged and organized tasks such as completing tasks and projects to achieve a specific goal such as passing exams. Thus, the term engagement can be extended to include the school.

The model of engagement extracted by Schaufeli et al. [21] comprises three dimensions: vigor, dedication, and absorption. Vigor refers to high levels of energy and mental resilience while working, the willingness to invest effort in one's work, and persistence even in the face of difficulties. Dedication is characterized by a sense of significance, enthusiasm, inspiration, pride, and challenge. Absorption is characterized by being fully concentrated and deeply engrossed in one's work. These three dimensions are separate and yet interconnected and complementary in the school context [25].

In the study by Sengsouliya, et al. [26], students had a high level of engagement in terms of the emotional, behavioral, and cognitive dimensions. They were more engaged in learning when teachers provided them with opportunities for discussions with their peers. Shih [27] reported that teacher support during the school day had a positive impact on students' AE in all three behavioral, cognitive, and emotional aspects, and thus had a significant impact on academic achievement. Buzzai, et al. [28] found that feelings of satisfaction and academic achievement are positively related to AE. A survey of literature reveals a number of factors that are closely related to AE, e.g., motivation, colleagues, teachers, and family [26]. AE with its three dimensions is also linked to the academic environment.

Among the important previous studies that addressed AE is Al-Mishmishi's study [29] that aimed to identify the contribution of problem-solving skills to predicting AE among students of the College of Education. The study followed the descriptive approach, where data collection tools were applied to 263 science and art students. The study found a positive correlation between AE and problem-solving skills. Students' AE could be predicted by problem-solving skills. The study of Zaki and Salem (2023) investigated the relative contribution of regulatory focus, academic hardness, and implicit intelligence to online learning engagement among Benha University students. The study followed the descriptive approach, where data was collected via questionnaires from 92 students. The study found a positive correlation between commitment and behavioral engagement in online learning [5].

Ali [9] aimed to reveal the possibility of understanding and interpreting the best structural model for explaining the direct and indirect effects of the dimensions of positive academic emotions on AE through academic grit as a mediating variable among a sample of 587 first year secondary school students. The results revealed a good fit of the proposed structural model of the relationships among all dimensions of positive academic emotions and AE.

Al-Hadhali and Al-Harbi [30] examined the relationship between mental wandering and AE. The study sample consisted of 420 bachelor degree students at Umm Al-Qura University. The AE scale developed by Mahasneh et al. [31] was applied to the participants. The results showed that Umm Al-Qura University students had a high level of AE. The study recommended training students on identifying their academic problems and solving them, which can contribute to increasing AE.

Al-Qasabi [32] explored the effect of short-term targeted attention training in reducing mental wandering during e-learning among university students. The final study sample consisted of 52 university students who were randomly and equally assigned to experimental and control groups. The results supported the effectiveness of attention training in reducing mental wandering in the experimental group.

Al-Harbi [33] sought to reveal the differences between the hypothesized mean of AE and its sub-dimensions and the actual mean among Qassim University students. A cohort of 400 female students were administered the AE questionnaire developed by Al-Janadi and Talab [34] The study found statistically significant differences between the respondents' mean scores on the AE questionnaire in favor of the actual mean.

Abbas [35] studied AE among university students and differences in it by gender and specialization. The study sample consisted of 376 students who responded to the AE questionnaire developed Abu Qura [36]. The participants were found to have high AE. No statistically significant differences in AE by gender or specialization were found.

Finally, Al-Najjar [37] sought to identify the relationship between mental alertness and both the need for knowledge and AE among graduate students at the College of Education (N = 296). Another aim was to identify the possibility of predicting AE by mental alertness and the need for knowledge. The study found a positive correlation between some dimensions of mental alertness and the need for knowledge, and some dimensions of AE.

**The research context**

Population of the study are artistically gifted students from programs in Prince Sattam bin Abdulaziz University other than fine arts (medicine, engineering, science, business administration, etc). They are admitted to the Gifted Care Unit based on a practical test and personal interviews. The unit provides them with qualifying courses in a number of arts and the tools they need while completing their artworks. At the beginning of the academic year, the Deanship of

Student Affairs launches a university-wide talent competition, including fine arts. A committee of five faculty members in the arts department evaluates artistically gifted applicants by a practical test and personal interviews. Students meeting the selection criteria are then registered in the Gifted Care Unit according to the arts they are gifted in, e.g., drawing, sculpture, graphic design, and photography. At their convenience, the students keep attending at the unit to produce their artworks, and from time to time they are provided with specialized training. They also receive guidance from the specialists at the unit.

## Method

**Study design.** This exploratory study followed a mixed-methods design involving quantitative and qualitative data collection and analysis. The participating students responded to a 30-item author-developed questionnaire assessing their ASs and AE using a five-point Likert scale ranging from 1 "strongly disagree" to 5 "strongly agree". Data obtained from responses to these 30 items was treated quantitatively and used to answer the first three research questions. The questionnaire also had an open-ended question about ways to promote students' ASs. The participants' answers to this question were analyzed qualitatively to extract common themes. Furthermore, structured interviews were conducted with members of the university's Gifted Care Unit Council on the basis of voluntary participation. Those members were asked the same open-ended question in the students' questionnaire about ways to promote students' ASs. Data obtained from the interviews were qualitatively analyzed for common themes. Members' feedback together with students' responses answered the fourth research question about ways to promote students' ASs.

## Participants

After receiving ethical approval from the Standing Committee for Bioethics Research (SCBR) at Prince Sattam University No. 307/2024, on 04/07/2024, data collection, including questionnaire distribution and interviews, began on 10/07/2024, and was completed on 22/08/2024. A cohort of 102 male and female artistically gifted non-specialist students enrolled in the Plastic Arts Section of the Gifted Care Unit at Prince Sattam bin Abdulaziz University, Saudi Arabia responded to the study questionnaire probing ASs and AE. Their ages ranged from 18 to 23 years. They all came from undergraduate programs in theoretical and applied colleges: 37% from applied colleges, 33% from theoretical colleges, 15% from health colleges, 13% from the diploma program, and 4% from the preparatory year. Student participation was optional. An electronic link of the questionnaire was sent to the students who were informed that they had the choice to complete the questionnaire or decline it. Thus, their completion of the questionnaire implied their consent to participate. Furthermore, 14 members of the university's Gifted Care Unit participated in structured interviews to probe their opinions on how to promote students' ASs. Those members varied in terms of gender, qualification, and experience.

## Instruments

**The questionnaire.** A 30-item questionnaire was developed by the author to probe students' ASs and AE. The construction of the questionnaire began with a literature review of tools assessing ASs [4,14] and AE [6,22,38,39]. The preliminary version of the questionnaire that had 35 items was face-validated by a number of specialists who were invited to judge the wording of the items and the inclusion of items under their respective dimensions. Based on their feedback, some items were reworded and some other items (N = 5) were excluded, leaving the questionnaire with 30 items. The final version of the questionnaire had two subscales: ASs (17) items and AE (13 items). The ASs subscale had 3 dimensions: cognitive skills (6 items), performance skills (6 items), and emotional skills (5 items). The AE subscale also had three dimensions: academic vigor (5 items), academic dedication (4 items), and academic absorption (4 items). The students responded to the questionnaire items based on a five-point Likert scale ranging from 1 "strongly disagree" to 5 "strongly

agree". Additionally, the questionnaire had an open-ended question about ways to promote students' ASs. The students were asked to provide their suggestions in response to this question.

After obtaining the ethical approval, the questionnaire was electronically administered to the students. The obtained data was then statistically treated to establish the questionnaire's validity and reliability. To check the construct validity of the questionnaire, correlations among items and their respective dimensions and among items and the total score were calculated. Items correlated with their respective dimensions with coefficients ranging between 0.53 and 0.85 and with the total score with coefficients ranging between 0.30 and 0.79. All correlation coefficients were statistically significant at the 0.01 level, indicating that the questionnaire was internally consistent. These results are shown in Table 1. The questionnaire's reliability was checked by the Alpha Cronbach method. The alpha estimates of the dimensions and the total questionnaire ranged between 0.67 and 0.90, indicating that the questionnaire was quite reliable. These results are listed in Table 2.

**The interview.** The author conducted structured interviews with 14 members of the Gifted Care Unit. They were asked the same question included in the students' questionnaire about ways to promote students' ASs. In the interviews, the author would ask probing questions to obtain further explanations from the interviewees. The interviews were conducted electronically and recorded with the interviewees' consent. The recorded interviews were later analyzed for common themes.

**Table 1. Correlations among items, dimensions, and the total score.**

| Item | Cor. with dimension | Cor. with the total score | Item | Cor. with dimension | Cor. with the total score | Item | Cor. with dimension | Cor. with the total score |
|------|------|------|------|------|------|------|------|------|
| Artistic Skills | | | | | | | | |
| 1 | 0.83** | 0.55** | 7 | 0.54** | 0.49** | 13 | 0.81** | 0.55** |
| 2 | 0.80** | 0.62** | 8 | 0.62** | 0.51** | 14 | 0.71** | 0.44** |
| 3 | 0.85** | 0.70** | 9 | 0.63** | 0.55** | 15 | 0.78** | 0.55** |
| 4 | 0.81** | 0.64** | 10 | 0.68** | 0.60** | 16 | 0.79** | 0.50** |
| 5 | 0.69** | 0.66** | 11 | 0.53** | 0.30** | 17 | 0.79** | 0.30** |
| 6 | 0.80** | 0.57** | 12 | 0.69** | 0.49** | | | |
| Academic Engagement | | | | | | | | |
| 1 | 0.75** | 0.49** | 6 | 0.70** | 0.51** | 10 | 0.67** | 0.48** |
| 2 | 0.73** | 0.39** | 7 | 0.79** | 0.43** | 11 | 0.76** | 0.51** |
| 3 | 0.80** | 0.50** | 8 | 0.83** | 0.50** | 12 | 0.73** | 0.49** |
| 4 | 0.74** | 0.43** | 9 | 0.69** | 0.37** | 13 | 0.77** | 0.51** |
| 5 | 0.54** | 0.45** | | | | | | |

** Significant at the 0.01 level.

**Table 2. Alpha coefficients for the questionnaire's reliability.**

| No. | Variable | Alpha Coefficient |
|-----|----------|-------------------|
| 1 | Cognitive skills | 0.88 |
| 2 | Performance skills | 0.67 |
| 3 | Emotional skills | 0.83 |
| 4 | Academic vigor | 0.76 |
| 5 | Academic dedication | 0.74 |
| 6 | Academic absorption | 0.71 |
| 7 | The total questionnaire | 0.90 |

## Data analysis

The quantitative data analysis was performed using the Statistical Package for the Social Sciences (SPSS) Version 22. Descriptive statistics were used to identify the level of students' ASs and AE. Pearson correlation coefficients were also used to identify the correlations among the study variables. Stepwise multiple regression analysis was performed to identify the prediction of students' ASs by their AE. Responses given by students and members of the Gifted Care Unit to the open-ended question were analyzed qualitatively for common themes.

## Results

### The level of students' ASs and AE

Prior to statistical analysis, the normal distribution of scores was examined by the Kolmogorov-Smirnov Test. All the Kolmogorov-Smirnov Test values were greater than 0.05, indicating that all sets of scores were normally distributed. To consider a mean as high, medium, or low, Oxford's [40] scoring system was used: high (mean of 3.5 or higher), medium (mean of 2.5–3.4), and low (mean of 2.4 or lower).

### Artistic skills

To identify the students' level of ASs, means and standard deviations were calculated. These results are presented in Table 3.

Table 3. The descriptive statistics of students' ASs in descending order.

| No. | Items | M | SD | Degree of Agreement |
|---|---|---|---|---|
|  | Cognitive Skills |  |  |  |
| 1 | I link the content of artistic works to life and contemporary events in society. | 4.06 | .931 | High |
| 2 | I have the ability to compare works of art. | 4.01 | .895 | High |
| 3 | I am familiar with artistic concepts. | 3.91 | .924 | High |
| 4 | I have knowledge of the development of art throughout history. | 3.74 | 1.05 | High |
| 5 | I have the ability to analyze artworks. | 3.73 | 1.06 | High |
| 6 | I follow developments in the field of plastic arts. | 3.56 | 1.18 | High |
|  | Total | 3.84 | .805 | High |
|  | Performance Skills |  |  |  |
| 1 | I have the skill of drawing with a pencil. | 4.02 | 1.12 | High |
| 2 | I use modern artistic techniques when implementing my artistic works. | 3.87 | 1.05 | High |
| 3 | I have skill in painting using colors. | 3.84 | 1.05 | High |
| 4 | I abstain from imitating others while producing artistic works. | 3.81 | .876 | High |
| 5 | I make use of appropriate environmental materials to produce my artworks. | 3.79 | 1.06 | High |
| 6 | I improve my experience by attending workshops and training courses in the arts. | 3.16 | 1.23 | Medium |
|  | Total | 3.75 | .659 | High |
|  | Emotional Skills |  |  |  |
| 1 | I appreciate works of art from different cultures. | 4.63 | .716 | High |
| 2 | I appreciate artworks in exhibitions. | 4.45 | .816 | High |
| 3 | I can taste the beauty of works of art. | 4.37 | .911 | High |
| 4 | I realize the aesthetic dimensions of the colors used in the artwork. | 4.31 | .867 | High |
| 5 | I express my feelings through my art practice. | 4.22 | .953 | High |
|  | Total | 4.40 | .664 | High |

Data in Table 3 reveals that the means of the three dimensions of ASs are high. Emotional skills ranked first with a mean of 4.40, followed by cognitive skills (M = 3.84) and performance skills (M = 3.75. But for one item, all items achieved high means. The only item that achieved a medium agreement is "I improve my experience by attending workshops and training courses in fine arts". This may hint to some kind of weakness in the training provided to artistically gifted students either because students are not encouraged to attend training events or because not much training is offered to them. Overall, these results reveal that the students' ASs are high.

## Academic engagement

Table 4 below presents the descriptive statistics of the students' AE.

As listed in Table 4, the means of the three dimensions of AE are high. Academic dedication achieved the highest mean (M = 4.23), followed by academic vigor (M = 3.83) and academic absorption (M = 3.61). But for two items, the means of all other items are high. Overall, these results indicate a high level of AE among students.

## The relationship between ASs and AE

To identify the relationship between academic skills and AE, Pearson correlations were calculated. These results are listed in Table 5 below.

As listed in Table 5, the strongest correlations are among dimensions of each of the two constructs, i.e., ASs and AE. This supports the construct validity of the questionnaire. As to the relationship between ASs and AE, cognitive ASs correlated positively and significantly with academic vigor (r = 0.22) and academic absorption (r = 0.29). However, their correlation with academic dedication is positive but insignificant (r = 0.19). Performance ASs correlated positively and significantly with all dimensions of AE: academic vigor (r = 0.30), academic dedication (r = 0.30), and academic absorption

**Table 4. The descriptive statistics of the students' AE in descending order.**

| No. | Items | M | SD | Degree of Agreement |
|---|---|---|---|---|
| | Academic Vigor | | | |
| 1 | I persevere in my studies even when things are not going well. | 4.19 | .829 | High |
| 2 | I have great flexibility in my studies. | 3.82 | .905 | High |
| 3 | I have the desire to study when I wake up in the morning. | 3.76 | 1.03 | High |
| 4 | I feel active and energetic while studying. | 3.73 | .911 | High |
| 5 | I have the ability to study for long hours. | 3.65 | 1.03 | High |
| | Total | 3.83 | .676 | High |
| | Academic Dedication | | | |
| 1 | I am proud of my field of study. | 4.55 | .852 | High |
| 2 | I try to overcome obstacles during my studies. | 4.34 | .724 | High |
| 3 | I feel enthusiastic about my studies. | 4.11 | .831 | High |
| 4 | I achieve my goals by studying. | 3.94 | 1.04 | High |
| | Total | 4.23 | .654 | High |
| | Academic Absorption | | | |
| 1 | I feel comfortable while studying my lessons. | 3.95 | .958 | High |
| 2 | Time passes quickly when I'm studying. | 3.70 | 1.08 | High |
| 3 | I feel happy when I'm studying intensively. | 3.43 | 1.25 | Medium |
| 4 | I forget everything around me when I'm studying. | 3.35 | 1.03 | Medium |
| | Total | 3.61 | .791 | High |

(r = 0.33). Emotional ASs correlated positively and significantly with academic dedication (r = 0.23) and academic absorption (r = 0.22). Their correlation with academic vigor was positive but insignificant (r = 0.12).

The relationship between the total ASs and AE is positive and significant (r = 33) as shown in Table 6 below. Overall, these results indicate that students with higher levels of ASs are more academically engaged than their counterparts with lower ASs.

To further evaluate the contribution of ASs to AE, stepwise regression analysis was conducted with the three individual dimensions of ASs and total ASs as predictor variables and total AE as a dependent variable. Multicollinearity was first checked to identify if the stepwise method could be used. No multicollinearity issue was detected, as correlations among the independent variables were not high. Furthermore, the VIF values for all the variables ranged from 1.1 to 2.2, which are all acceptable. The analysis produced one model where only performance ASs positively and significantly predicted AE. The regression model is shown in Table 7.

It is clear from Table 7 that performance ASs significantly and positively predicted students' AE (β = 0.36, t = 3.9, p ≤ 0.001). This indicates that students' performance ASs explained 13% of variance in their AE. That is, of the three dimensions of AS, performance skills proved to be the only predictor of AE.

When standard regression was conducted with total ASs as a predictor variable and total AE as a dependent variable, the extracted model (Table 8) was significant. AS significantly and positively predicted AE (β = 0.36, t = 3.9, p ≤ 0.001). ASs explained 11% of variance in AE. That is, students with higher level of ASs are more academically engaged.

**Table 5.  Correlations among student's ASs and AE.**

|  | 1 | 2 | 3 | 4 | 5 | 6 |
|---|---|---|---|---|---|---|
| 1. Cognitive skills | – |  |  |  |  |  |
| 2. Performance skills | 0.70** | – |  |  |  |  |
| 3. Emotional skills | 0.60** | 0.57** | – |  |  |  |
| 4. Academic vigor | 0.22* | 0.30** | 0.12 | – |  |  |
| 5. Academic dedication | 0.19 | 0.30** | 0.23* | 0.56** | – |  |
| 6. Academic absorption | 0.29** | 0.33** | 0.22* | 0.63** | 0.54** | – |

**Table 6.  The relationship between the total and AE.**

|  |  | Artistic skills | Academic engagement |
|---|---|---|---|
| Artistic skills | Pearson Correlation | 1 | 0.33** |
|  | Sig. (2-tailed) |  | 0.001 |
|  | N | 102 | 102 |
| Academic engagement | Pearson Correlation | 0.33** | 1 |
|  | Sig. (2-tailed) | 0.001 |  |
|  | N | 102 | 102 |

**Table 7.  Stepwise multiple regression for predicting AE by AS.**

| Predictors | R | R² | F | B | SE | β | t | p |
|---|---|---|---|---|---|---|---|---|
| Performance skills | 0.36 | 0.132 | 15.2*** | 0.33 | 0.085 | 0.36 | 3.9 | .000 |

**Table 8. Standard regression for predicting AE by AS.**

| Predictors | R | R² | F | B | SE | β | t | p |
|---|---|---|---|---|---|---|---|---|
| Ass | 0.33 | 0.111 | 12.4** | 0.32 | 0.091 | 0.33 | 3.5 | .001 |

## Ways to promote students' ASs

To answer the research question about ways to promote students' ASs, the participating students' responses to the open-ended question and the feedback given by members of the Gifted Care Unit in the interview were qualitatively analyzed.

### Students' feedback

The researcher read the students' responses to the open-ended question about ways to promote their ASs. The students offered 12 themes, the strongest of which was offered by 56% of the students, while the weakest was offered by 6% of the students. The following table summarizes the themes with sizable agreement from the students.

As listed in Table 9, the suggestions provided by the largest number of students centered on the provision of specialized training, supportive environment, and financial and spiritual support. The other highly agreed upon suggestions centered on attending art galleries and interaction with professional artists for experience sharing. A sizeable number of students also suggested continuous practice of the ASs. It is clear that most of the suggestions offered are the responsibility of the university, and few of them need personal initiative.

### Members' feedback

A thematic analysis [41] was conducted to investigate the open-ended responses gathered from the survey instrument. The researcher developed a coding scheme to systematically classify the data and identify emergent trends. The analytical framework was subsequently applied to the interview transcripts, following the same methodological procedures. Thematic patterns were identified through iterative reading, coding, and data refinement. A comparative analysis was performed to evaluate the consistency and discrepancies between the themes extracted from the survey responses and those from the interview data, subsequent to the formulation of a cohesive set of themes from both sources. Where subject convergence was identified, integration was implemented to enhance the interpretive depth of the findings. All definitive topics were examined and validated by the researcher to ensure analytical rigor.

**Table 9. Summary of the themes emerged from students' responses.**

| No. | Themes | % |
|---|---|---|
| 1 | Attending training workshops and courses | 56 |
| 2 | Continuous practice of the AS | 53 |
| 3 | Visiting art galleries | 51 |
| 4 | Provision of a supportive environment for performing various artistic works by the university | 47 |
| 5 | Provision of financial and spiritual support to artistically gifted students by the university and the family | 46 |
| 6 | Communication and interaction with professional artists | 38 |
| 7 | Provision of spaces to practice AS | 29 |
| 8 | Art competitions | 26 |
| 9 | Participation in group artistic activities | 18 |

**The role of the gifted care unit.** All the members pointed out the important role the unit plays to nurture artistically gifted students and to spread the art culture throughout the university. They confirmed that the unit launches enrichment programs in fine arts, many of which target educating members of the unit and interested faculty members in the university about the best ways to nurture students' ASs. They mentioned examples of the training courses provided by the unit, e.g., a training course titled "Gifted people: Investing in generations of scholars and geniuses". When asked the question, member 3 stated, "The unit focuses on teaching technical skills and expanding students' creative horizons by connecting them with diverse cultural and artistic topics. This contributes to shaping a comprehensive artistic vision among students, thus enhancing the university's role in spreading artistic culture widely." When the same question was asked to member 14, he said, "Through the activities organized by the unit, students are able to develop their technical skills and enhance their creative sense. The activities provided by the unit are not limited to learning technical skills, but also include employing these skills in the labor market, which helps students produce distinctive, relevant artwork."

**The need for more training and identification efforts.** Eleven of the 14 members suggested that artistically gifted students need more specialized training opportunities. They mentioned areas where students need training, e.g., acrylic coloring techniques, the basics of the graphic design, and forming jewelry with metal minting. Nine of the members (64%) pointed out the need to launch identification expeditions to identify artistically gifted students from all the colleges to reach neglected artistically gifted students. Member 1 expressed a clear opinion on the need to increase training efforts directed at artistically gifted students. He stated that it is necessary to introduce new technologies into training, such as training in the arts of photography and video art, as well as the need to emphasize training on modern tools used in digital arts. He added: "Technological development has added new dimensions to art, and this must be reflected in the training programs we provide to our students, which aim to gradually develop their skills, especially in the arts of digital photography."

**The need to link students to outside sources of experience.** Eight members confirmed that artistically gifted students need to be linked to outside sources of artistic experience. For this reason, they mentioned that students need to pay more frequent visits to art galleries inside and outside the Kingdom of Saudi Arabia in the company of supervisors from the unit. They also recommended that professional national and international artists be invited to the unit to share their experience with students. As they commented, the unit actually invites professional artists, but not with a satisfactory frequency. In this respect, the head of the unit stated, "We need and plan to send our students to more art galleries because galleries stimulate students' creativity and artistic production". Member 9 said, "We need to increase the connection between students and the global art community. The Gifted and Talented Unit could organize periodic visits to international exhibitions or meetings with well-known artists to broaden students' horizons and develop their artistic culture."

Member 5 also stated, "Students should be more exposed to direct contact with art". Therefore, he suggested that there be traveling exhibitions on campus that display students' works periodically, with prizes allocated to encourage the winners." He continued, "Traveling exhibitions give students the opportunity to see their own work alongside that of their peers, which helps them develop their artistic taste and increase their creativity." Regarding meetings with artists, member 10 emphasized the need to organize periodic visits to some artists in various artistic fields saying: "Exposure to diverse artistic cultures and organized dialogues through students' meetings with professional artists will contribute to enriching the students' artistic experience and expanding their horizons, and this will give them a new vision that will help them innovate and create in their artistic works."

**The need for family support.** Eight members confirmed that families can play a significant role in the nurturing of their children's artistic gifts. Member 8 stated, "Significant others, especially family members, should provide artistically gifted students with financial and more importantly spiritual support. This is why the unit should be in touch with families and educate them about how to encourage their children to take their talents to higher levels". Member 11 also pointed out that "family support is a pivotal factor in developing students' artistic skills, as it encourages them to continue their artistic career." He added that it is important to organize workshops for families to educate them on how to encourage their children to develop their artistic skills.

**The need to stimulate students' creativity.** Seven members raised the need for creating the spirit of completion among artistically gifted students to stimulate them to be more creative in their art productions. They therefore suggested holding more art competitions, propagandizing them inside and outside the university, and allocating tempting prizes for winners. Member 11 stated, "Art competitions are very stimulating to students' creativity. We need to organize more art competitions and invite professional artists to referee students' productions". Member 6 proposed that "adding a motivational aspect through artistic challenges could encourage students to work on innovative projects, as it would be useful to organize artistic competitions, where students compete to implement innovative artistic projects, with prizes awarded to encourage the spirit of competition and creativity."

**Triangulation of quantitative and qualitative data.** Triangulation [39] was conducted to improve the validity of the findings by comparing and integrating results from the quantitative and qualitative components of the study. The researcher employed analytical worksheets to organize and cross-reference the datasets, enabling the identification of convergent and divergent trends. The integrated analysis was reported in accordance with the Good Reporting of a Mixed Methods Study checklist to ensure transparency and methodological rigor. Table 10 presents a summary of the triangulated findings.

## Discussion

The results revealed that students' cognitive ASs are high. This can be attributed to the Gifted Care Unit's efforts in providing students with artistic experiences and knowledge. Furthermore, artistically gifted students may have an inherent interest in being updated on the developments in their preferred arts. One more possible explanation is the ease of accessing art knowledge thanks to the ever-expanding information technology. Information technology and devices are readily available in Saudi Arabia, and because of being well-off, Saudi students possess the latest technological devices,

**Table 10. Triangulation of quantitative and qualitative data.**

| Themes for Quantitative data | Themes for Qualitative data | Sub-Themes |
|---|---|---|
| ____ | The Role of the Gifted Care Unit | Nurturing artistically gifted students and spreading the art culture throughout the university. |
| | | Launching enrichment programs in fine arts |
| Attending training workshops and courses | The Need for More Training and Identification Efforts | Acrylic coloring techniques |
| | | The basics of the graphic design |
| Provision of spaces to practice AS | | Forming jewelry with metal minting. |
| Continuous practice of the AS | | |
| Visiting art galleries | The Need to Link Students to Outside Sources of Experience | The need to pay more frequent visits to art galleries inside and outside the Kingdom of Saudi Arabia |
| Communication and interaction with professional artists | | Professional national and international artists be invited to the unit to share their experience with students. |
| Provision of financial and spiritual support to artistically gifted students by the university and the family | The Need for Family Support | Providing artistically gifted students with financial |
| | | Spiritual support. |
| | | Holding more art competitions |
| | | Propagandizing them inside and outside the university, |
| Provision of a supportive environment for performing various artistic works by the university | The Need to Stimulate Students' Creativity | Allocating tempting prizes for winners |
| Participation in group artistic activities | | |
| Art competitions | | |

which they can use efficiently. This finding is in line with the studies of [16,17], where students' were found to have sophisticated artistic knowledge owing to the availability of technological information sources. The students' possessed high performance ASs. This may refer to the effective nomination and admission of gifted students to the Gifted Care Unit. One other possible reason is the specialized training and support provided to students by the unit. The interview with members of the unit revealed that the unit provides students with workshops to promote their performance. This finding converges with Salama's study [17], which found that providing students with specialized training strengthens their ASs. This finding is also in line with the studies of Saleh and Al-Bashir [19,20], which concluded that students could promote their performance thanks to the availability of tools and modern art technologies. As to emotional ASs, the students were found to have highly positive artistic attitudes. This seems logical because art is their passion. This finding is consistent with several previous studies, e.g., the studies of Ahmed [11,15,18], where artistically gifted students were reported to be highly appreciative of artistic expression.

As to AE, the students in this study were found to be highly academically engaged in terms of academic vigor, dedication, and absorption. ASs give students the feelings of satisfaction and pride. One may allege that ASs flourish the spirit of serenity in students. They make their imagination vivid. All these attributes may contribute to artistically gifted students' being focused on their academic study. This same finding was reported by previous studies [7,38,35], where comparable samples of students were found to be more academically engaged than students without ASs. Taha [38] concluded that students possessing the feelings of satisfaction, happiness, optimism, and pride are more academically engaged than students lacking these feelings. Practicing arts can provide students with these feelings.

More importantly, this study reported a positive and significant correlation between ASs and AE. ASs as a whole were found to explain 11% of variance in students' AE. This suggests that pursuing arts fosters students' AE. Of all the dimensions of ASs, only performance skills predicted students' AE. They explained 13% of the variance in students' AE. This finding seems reasonable since performance is the executive aspect of art and is therefore more influential on students' mentality. Al-Zahrani [10] reported that students' participation in extracurricular activities fosters students' AE. Of course ASs come at the top of extracurricular activities in the University of this Study' population. Knoop [42] suggested that activities that give students satisfaction, happiness, optimism, hope, and pride foster their motivation, attention, and self-regulated learning. In this same respect, Suleiman [43] found that pursuing art enhances students' mental abilities.

The author thinks that practicing ASs helps students get rid of negative emotions as confirmed by the studies of Al-Arabi [44]. Rivera [45] suggests that art is important for students who are unable to express themselves in words or communicate verbally. Art enables them to express their emotions, thoughts, fears, and fancies. Diehls [46] also believes that art gives students the opportunity to express their feelings and thoughts through visual images such as drawings, photographs, and sculptures. This makes them feel relaxed and reduces their anxiety, which makes them more prepared to express themselves and release their inner feelings.

Analysis of students' responses to the open-ended question about ways to promote their ASs and the feedback provided by members of the Gifted Care Unit in the interviews revealed several suggestions that both samples agreed upon. These included, among others, the need to (1) provide students with more specialized training, (2) link them to outside sources of artistic experience, e.g., art galleries, (3) provide them with support, especially spiritual support from significant others, and encourage them to be more creative in their artistic production through competitions.

## Conclusions

The important finding in this study is that ASs, especially performance skills, contributes to students AE. This makes it important for specialized entities like the Gifted Care Unit to exert more efforts to identify artistically gifted students in all university programs. These efforts can identify students who are aware that they have artistic skills and students who have artistic skills but are unaware of them. The art culture should be spread all over the university to attract more students to join the Gifted Care Unit. The artistically gifted students should be provided with specialized training, contact with

outside sources of artistic experience, an art-nurturing environment, incentives to be more creative, and support, especially spiritual support.

## Research limitations

The generalizability of this study's findings is limited by the somehow small size of the sample. Future investigations can yield more generalizable results by using a larger sample. It is also recommended that future investigations include interviews with students to elicit from them more illuminating information.

## Supporting information

**S1 Data. The questionnaire.**
(DOCX)

**S2 Data. Raw data.**
(XLSX)

**S3 Data. Quantified data.**
(XLSX)

**S4 Data. Data analysis**
(XLSX)

## Author contributions

**Conceptualization:** Ghozeail Abdulaziz Abdullah Aldhorman.

**Data curation:** Ghozeail Abdulaziz Abdullah Aldhorman.

**Formal analysis:** Ghozeail Abdulaziz Abdullah Aldhorman.

**Funding acquisition:** Ghozeail Abdulaziz Abdullah Aldhorman.

**Investigation:** Ghozeail Abdulaziz Abdullah Aldhorman.

**Methodology:** Ghozeail Abdulaziz Abdullah Aldhorman.

**Project administration:** Ghozeail Abdulaziz Abdullah Aldhorman.

**Resources:** Ghozeail Abdulaziz Abdullah Aldhorman.

**Software:** Ghozeail Abdulaziz Abdullah Aldhorman.

**Supervision:** Ghozeail Abdulaziz Abdullah Aldhorman.

**Validation:** Ghozeail Abdulaziz Abdullah Aldhorman.

**Visualization:** Ghozeail Abdulaziz Abdullah Aldhorman.

**Writing – original draft:** Ghozeail Abdulaziz Abdullah Aldhorman.

**Writing – review & editing:** Ghozeail Abdulaziz Abdullah Aldhorman.

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
