## [Decision Letter · Decision Letter 0]

16 Apr 2025

PONE-D-24-59201The Relationship between Artistic Skills and Academic Engagement among Artistically Gifted Non-Specialist University StudentsPLOS ONE

Dear Dr. Aldhorman,

Thank you for submitting your manuscript to PLOS ONE. After careful consideration, we feel that it has merit but does not fully meet PLOS ONE’s publication criteria as it currently stands. Therefore, we invite you to submit a revised version of the manuscript that addresses the points raised during the review process.

We look forward to receiving your revised manuscript.

Kind regards,

Ruth Nayibe Cardenas Soler, Ph.D.

Academic Editor

PLOS ONE

Journal Requirements:

3. We note that your Data Availability Statement is currently as follows: All relevant data are within the manuscript and in Supporting Information files.

Additional Editor Comments:

Please review the reviewers' comments and make any suggested corrections.

Reviewers' comments:

Reviewer's Responses to Questions

**Comments to the Author**

1. Is the manuscript technically sound, and do the data support the conclusions?

Reviewer #1: Partly

Reviewer #2: Yes

2. Has the statistical analysis been performed appropriately and rigorously? 

Reviewer #1: Yes

Reviewer #2: No

3. Have the authors made all data underlying the findings in their manuscript fully available?

Reviewer #1: Yes

Reviewer #2: Yes

4. Is the manuscript presented in an intelligible fashion and written in standard English?

Reviewer #1: No

Reviewer #2: Yes

5. Review Comments to the Author

Reviewer #1: Dear Author,

Thank you for the opportunity to review this manuscript. Below are the comments on my review:

1. The subject matter of the article is relevant to be published in the journal. However, it needs some revisions.

2. The title clearly reflects the subject matter of the article.

3. The abstract and keywords provide good information about the article. The subject matter, the objective of the research, the methodology used, the main results and conclusions are mentioned.

4. Check whether all authors cited in the article are in the manuscript’s references, and vice versa.

5. The article has problems with clarity and coherence in its language, which need to be revised. In the Introduction, it is important to make clear what the objective of the research is, and the reasons that led to the development of this study. In the methodology, it is necessary to make clear some points:

- What was the same question asked to the students and to the members of the university’s Gifted Care Unit Council? Was it appropriate to ask the same question? Make it clear which colleges the participating students attended (this was confusing).

Regarding the conclusions, make it clear whether the questions asked were answered. It is also important to mention the relevance of the research developed in the conclusion.

6. There is articulation between the theme and the theoretical basis. There is an interesting dialogue with the scientific literature in the area, but this discussion could be further explored.

7. There is analysis of the data and coherence in the argument. The data is organized, categorized, and analyzed. Something stands out: in the students' feedback regarding group artistic activities, only 18% responded about ways to promote their ASs. It is important to analyze this, because the study covers artistic skills and academic engagement. What does the scientific literature say about this? Could it be because some participating students did not come from arts colleges? (Make this information clearer in the methodology).

8. The bibliography used in the article is adequate and up-to-date.

It is an interesting study. However, I hope these comments help improve your manuscript.

Reviewer #2: 1. Research Value

Strengths:

The study addresses the relationship between artistic skills and academic engagement among non-specialist students with artistic talent. It adopts an interdisciplinary and policy-relevant perspective, offering practical implications for supporting cross-disciplinary artistic development in higher education institutions.

Suggestions:

The manuscript lacks a more thorough theoretical discussion and definition of “artistically gifted non-specialist students,” especially regarding selection and evaluation criteria. It is recommended to supplement the conceptual framework with references to talent identification systems.Clarify the novelty and theoretical contribution of this study compared to existing literature.

2. Methodology

Strengths:The use of a mixed-methods approach combining quantitative surveys and qualitative interviews is appropriate and strengthens the depth of the research. The questionnaire design is sound and informed by relevant literature, with appropriate construct validation.

Suggestions:

(1)More detail is needed regarding the validation of the questionnaire. Was exploratory or confirmatory factor analysis conducted?

(2)The sample is relatively limited (one university, one unit). Consider explicitly addressing this limitation and recommending broader sampling in future studies.

(3)The description of qualitative analysis is too brief. It is advisable to explain the coding process, theme extraction method, and whether triangulation or inter-coder reliability was used.

3. Data Analysis and Statistical Interpretation

Strengths: The use of descriptive statistics, Pearson correlation, and regression analysis is appropriate and generally well explained. The tripartite structure of artistic skills—cognitive, performance, and emotional—is useful for nuanced interpretation.

Suggestions:The regression models show low explanatory power (R² = 0.11–0.13). This should be more thoroughly discussed in terms of potential unmeasured mediators or moderators. The manuscript does not report whether multicollinearity was tested. Including VIF or tolerance values would enhance the robustness of the regression results.

4. Structure and Results Presentation

Strengths:Tables are clear and well-organized. Psychometric results (reliability and validity) are properly included. Integration of qualitative and quantitative results is smooth.

Suggestions:Some sections (Results and Discussion) are slightly repetitive. Consider refining to avoid restating the same results. Using visual aids such as conceptual diagrams or figures could improve clarity and enhance reader engagement.

5. Writing Quality

Strengths:The manuscript is generally well-written, logically structured, and linguistically competent. References are comprehensive and support the arguments.

Suggestions:Some phrases are overly wordy. For example, “students who are aware that they have artistic skills and students who have artistic skills but are unaware of them” could be simplified. A final round of proofreading is recommended to ensure conciseness and stylistic coherence.

6. PLOS authors have the option to publish the peer review history of their article (what does this mean? ). If published, this will include your full peer review and any attached files.

**Do you want your identity to be public for this peer review?** For information about this choice, including consent withdrawal, please see our Privacy Policy .

Reviewer #1: **Yes: ** Gustavo Cunha de Araujo

Reviewer #2: No

---

## [Author Response · Author response to Decision Letter 1]

11 Jun 2025

Response to academic editor and reviewers

The academic editor

Ensure that your manuscript meets PLOS ONE's style requirements

Addressed

Please note that funding information should not appear in any section or other areas of your manuscript.

Funding information was removed from the manuscript.

We note that your Data Availability Statement is currently as follows: All relevant data are within the manuscript and in Supporting Information files.

Please include captions for your Supporting Information files at the end of your manuscript, and update any in-text citations to match accordingly.

Data were provided as supporting information files.

Captions were provided at the end of the manuscript.

Review your reference list to ensure that it is complete and correct. Addressed

Documentation (in-text citations and references) was modified to be in accordance with the journal’s adopted system.

A repeated reference (No. 15) was removed and so numbers of references were changed in the text.

Reviewer 1

Check whether all authors cited in the article are in the manuscript’s references, and vice versa. Checked

The article has problems with clarity and coherence in its language, which need to be revised. In the Introduction, it is important to make clear what the objective of the research is, and the reasons that led to the development of this study. In the methodology, it is necessary to make clear some points:

- What was the same question asked to the students and to the members of the university’s Gifted Care Unit Council? Was it appropriate to ask the same question? Make it clear which colleges the participating students attended (this was confusing).

The article was checked for clarity and coherence.

The research objective was clarified in the introduction.

The open-ended question is mentioned in the abstract and the study design section: “The questionnaire also had an open-ended question about ways to promote students’ ASs”.

The same question had to be asked for the joint qualitative analysis of both samples’ responses. The researcher wished to identify the suggestions of both the students and the staff members regarding how to enhance students’ talents.

Students’ colleges are mentioned in the participants section.

Reviewer 2

The manuscript lacks a more thorough theoretical discussion and definition of “artistically gifted non-specialist students,” especially regarding selection and evaluation criteria. It is recommended to supplement the conceptual framework with references to talent identification systems. Clarify the novelty and theoretical contribution of this study compared to existing literature.

A section was added in the introduction with the heading “Research Context”. A more thorough discussion and definition of artistically gifted non-specialist students is provided in this section.

The description of qualitative analysis is too brief. It is advisable to explain the coding process, theme extraction method, and whether triangulation or inter-coder reliability was used. More details were added in the article.

As to the extraction method, the analysis was conducted by the researcher, given that the task was simple: what are the suggestions offered by students or staff members? It was a matter of how many students or staff members mentioned each suggestion.

The regression models show low explanatory power (R² = 0.11–0.13). This should be more thoroughly discussed in terms of potential unmeasured mediators or moderators. The manuscript does not report whether multicollinearity was tested. Including VIF or tolerance values would enhance the robustness of the regression results.

VIF values for the variables were added, which do not show any multicollinearity issues.

The description of qualitative analysis is too brief. It is advisable to explain the coding process, theme extraction method, and whether triangulation or inter-coder reliability was used. The coding process explained.

---

## [Editor Report · Decision Letter 1]

18 Jun 2025

PONE-D-24-59201R1The relationship between artistic skills and academic engagement among artistically gifted non-specialist university studentsPLOS ONE

Dear Dr. Aldhorman,

Thank you for submitting your manuscript to PLOS ONE. After careful consideration, we feel that it has merit but does not fully meet PLOS ONE’s publication criteria as it currently stands. Therefore, we invite you to submit a revised version of the manuscript that addresses the points raised by the reviewers during the review process.

We look forward to receiving your revised manuscript.

Kind regards,

Ruth Nayibe Cardenas Soler, Ph.D.

Academic Editor

PLOS ONE

Journal Requirements:

Additional Editor Comments :

The manuscript is suitable for publication but needs some minor adjustments. You have 45 days to revise and resubmit.

---

## [Author Response · Author response to Decision Letter 2]

14 Jul 2025

Comment: 1. It is important that you include a cover letter with your manuscript. Please ensure that this letter is addressed specifically to PLoS ONE. Please also include

* why this manuscript is suitable for publication in PLoS ONE.

* how does your paper provide a worthwhile addition to the scientific literature?

* how does your paper relate to previously published work?

* which types of scientists do you believe will be most interested in your study?

Response: I have added a cover letter as requested.

comment 2. We note that the grant information you provided in the ‘Funding Information’ and ‘Financial Disclosure’ sections do not match.

Response: I have fixed this issue.

---

## [Editor Report · Decision Letter 2]

17 Jul 2025

La relación entre las habilidades artísticas y el compromiso académico entre estudiantes universitarios no especialistas con talento artístico 

PONE-D-24-59201R2

Estimado Dr. Ghozeail Aldhorman

Nos complace informarle que su manuscrito ha sido evaluado científicamente como adecuado para su publicación y será aceptado formalmente para su publicación.

Se generará una factura cuando su artículo sea aceptado formalmente. Tenga en cuenta que, si su institución tiene un acuerdo editorial con PLOS y su artículo cumple con los criterios pertinentes, se cubrirán todos o parte de los costos de publicación. Asegúrese de que su información de usuario esté actualizada iniciando sesión en Editorial Manager®   y haciendo clic en el enlace "Actualizar mi información" en la parte superior de la página. Si tiene alguna pregunta sobre los cargos de publicación, comuníquese directamente con nuestro departamento de Facturación a Autores en authorbilling@plos.org.

Si su institución cuenta con una oficina de prensa, por favor, infórmeles sobre su próximo artículo para maximizar su impacto. Si van a preparar material de prensa, por favor, informe a nuestro equipo de prensa lo antes posible, en un plazo máximo de 48 horas tras recibir la aceptación formal. Su manuscrito permanecerá bajo estricto embargo de prensa hasta las 14:00 h (hora del este) del día de publicación. Para más información, por favor, contacte con onepress@plos.org.

Atentamente,

Ruth Nayibe Cardenas Soler, Ph.D. 

Editora académica 

de PLOS ONE

Comentarios adicionales del editor (opcional):

Comentarios de los revisores:

---

## [Editor Report · Acceptance letter]

PONE-D-24-59201R2

PLOS ONE

Dear Dr. Aldhorman,

I'm pleased to inform you that your manuscript has been deemed suitable for publication in PLOS ONE. Congratulations! Your manuscript is now being handed over to our production team.

Kind regards,

on behalf of

Dr. Ruth Nayibe Cardenas Soler

Academic Editor

PLOS ONE